# Hyaluronic Acid-Coated Nanomedicine for Targeted Cancer Therapy

**DOI:** 10.3390/pharmaceutics11070301

**Published:** 2019-06-30

**Authors:** Kibeom Kim, Huyeon Choi, Eun Seong Choi, Myoung-Hwan Park, Ja-Hyoung Ryu

**Affiliations:** 1Department of Chemistry, School of Natural Sciences, Ulsan National Institute of Science and Technology (UNIST), Ulsan 44919, Korea; 2Department of Chemistry and Life Science, Sahmyook University, Seoul 01795, Korea

**Keywords:** hyaluronic acid, cancer therapy, drug delivery, micelle, nanogel, silica nanoparticle, gold nanoparticle, metal organic framework

## Abstract

Hyaluronic acid (HA) has been widely investigated in cancer therapy due to its excellent characteristics. HA, which is a linear anionic polymer, has biocompatibility, biodegradability, non-immunogenicity, non-inflammatory, and non-toxicity properties. Various HA nanomedicines (i.e., micelles, nanogels, and nanoparticles) can be prepared easily using assembly and modification of its functional groups such as carboxy, hydroxy and *N*-acetyl groups. Nanometer-sized HA nanomedicines can selectively deliver drugs or other molecules into tumor sites via their enhanced permeability and retention (EPR) effect. In addition, HA can interact with overexpressed receptors in cancer cells such as cluster determinant 44 (CD44) and receptor for HA-mediated motility (RHAMM) and be degraded by a family of enzymes called hyaluronidase (HAdase) to release drugs or molecules. By interaction with receptors or degradation by enzymes inside cancer cells, HA nanomedicines allow enhanced targeting cancer therapy. In this article, recent studies about HA nanomedicines in drug delivery systems, photothermal therapy, photodynamic therapy, diagnostics (because of the high biocompatibility), colloidal stability, and cancer targeting are reviewed for strategies using micelles, nanogels, and inorganic nanoparticles.

## 1. Introduction

In 2018, 609,940 people in the United States died of cancer, one of the leading causes of death [1]. Chemotherapy is widely used to treat many types of cancer, but the low solubility of cancer therapeutics and the side effects caused by nonselective treatment remain a barrier. To overcome these problems, targeting nanomedicine is of interest, because it utilizes the anatomical and pathophysiological abnormalities, leukemic tumor vasculature, and overexpression of specific membrane proteins [2,3].

Hyaluronic acid (HA) is a linear anionic polymer composed of repeating disaccharide units of β-1,4-D-glucuronic acid-β-1,3-*N*-acetyl-D-glucosamine. It is a non-sulfated glycosaminoglycan found throughout the connective, epithelial, and neural tissues [4]. HA contains carboxylic acid, hydroxyl, and *N*-acetyl groups, and can easily be combined with other chemicals [5]. It is biocompatible, biodegradable, non-immunogenic, non-inflammatory, and non-toxic [6], and is widely used for arthritis treatment, ophthalmic surgery, drug delivery, and tissue engineering. As a drug delivery system, HA has been conjugated with various chemical drugs, such as paclitaxel (PTX) and doxorubicin (Dox), and other biopharmaceuticals [6]. The nanosized HA medicine (HA nanomedicine) is selectively transferred to cancer cells by the enhanced permeability and retention (EPR) effect [7,8,9]. In addition, selective delivery is enhanced by the reaction of HA with a variety of receptors, including cluster determinant 44 (CD44), receptor for HA-mediated motility (RHAMM), and lymphatic vessel endothelial receptor-1 (LYVE-1) [5,6,10]. HA nanomedicines generally have a negative surface charge, which prevents the clearance by the reticuloendothelial system (RES) [11]. For these reasons, HA nanomedicines have been extensively studied in the drug delivery field to increase the biocompatibility of the material and improve the delivery of drugs through passive and active targeting. In this review, we investigate the types, advantages, and therapeutic applications of HA nanomedicine (Figure 1).

## 2. HA Micelle 

Self-assembled polymeric micelles (PMs) have been extensively investigated as targeted drug carriers given their ability to solubilize water-insoluble anticancer drugs, as well as their excellent targeting activity to tumors [12,13,14]. PMs, which are amphiphilic, can encapsulate small hydrophobic drugs in their hydrophobic core via physical, chemical and electrostatic interaction [15,16,17], and can deliver hydrophobic drugs into their destinations [18]. Thus, PMs could improve the efficiency of anticancer drugs by enhancing drug solubility in water, prolonging blood circulation time, and selectively accumulating at the tumor via the EPR effect [19,20,21]. The nanosized (20 ~ 100 nm) structure of the PMs enables them to preferentially accumulate in angiogenic tumor tissues through the fenestrate vasculature following systemic administration [19,22]. HA can be conjugated with hydrophobic polymers, small molecules, and drugs to have high drug-loading efficiency. Additionally, HA can be coated with positively charged nanocarriers (Figure 2), and can achieve improved antitumor efficacy via releasing the drugs after specific stimuli, such as redox reactions or exposure to an acidic pH.

### 2.1. HA Polymeric Micelles (PMs)

Conjugating hydrophobic polymers via grafted polymer [23], block-co-polymers, and ring-opening polymers [24] with HA has been widely investigated to make amphiphilic polymers which can self-assemble into micelles [25]. These hydrophobic polymers include diverse hydrophobic units, such as caprolactone, methacrylate, and ethylene glycol. Park et al. presented a crosslinked HA PM, composed of polymers grafted to HA, such as HA-*b*-poly(caprolactone) (HA-*b*-PCL) and HA-poly(pyridyl disulfide methacrylate) (HA-*b*-P(PDSMA)) for in vivo studies [26,27,28].

In this crosslinking strategy, the disulfide bond was used to conjugate the 2-(pyridyldithio)-ethyl group on the methacrylate chain, which made a block-co-polymer with HA (Figure 3). Glutathione (GSH), which can reduce disulfide bonds, is abundant in the cytoplasm of cancer cells. During its initial circulation, the core-crosslinked HA PM has minimal leakage of the encapsulated drug; however, the GSH rich tumor-specific environment can trigger a burst release of drugs via cleaving the crosslinked disulfide bond. These bio-reducible PMs promote the release of drugs in specific stimuli to improve therapeutic efficacy. To improve selective accumulation at the tumor site, poly(ethylene glycol) (PEG)-conjugation (PEGylation) was developed [29,30]. The PEGylation provides several advantages, such as reduction of liver uptake and increased circulation time in the blood, but PEGylation can disrupt efficient interactions with receptors on the cancer cell [15]. 

The anticancer drug, camptothecin (CPT), was loaded into PEGylated HA micelles to enhance anticancer therapy. Hyaluronidase (HAdase) degrades CPT loaded HA nanoparticles (NPs), CPT-P-HA-NPs, which enables the release of CPT in tumor environment. The Costa group studied cancer chemo-photothermal therapy (PTT) with HA functionalized micelles loaded with Dox and IR780, which is a probe that is widely used for phototherapies [31]. Poly(maleic anhydride-*alt*-1-octadecene) (PMAO) was grafted onto HA to generate a hydrophobic moiety enabling self-assembly (HPN). This HA-*g*-PMAO can be loaded with IR780 and Dox for cancer chemo-phototherapy to make IR-HPN and IR/Dox-HPN.

Under near-infrared (NIR) irradiation, IR780 showed high cytotoxicity with PTT. Moreover, Dox loading provided chemotherapy and induced a synergistic effect in breast cancer cells. Thus, these strategies demonstrate that polymer grafted HA micelles have high cytotoxicity with drug loading and provide enhanced antitumor therapy in vitro and in vivo.

### 2.2. Small Molecules Conjugated to HA Micelles 

Instead of grafting hydrophobic polymers, small hydrophobic molecules can be conjugated to the *N*-acetyl group and primary hydroxyl group in HA [32]. Many hydrophobic drugs and small molecules have poor solubility in water. By combining water-soluble HA with these hydrophobic molecules, drugs can be delivered safely to their destination [33]. First, different amounts of a simple long alkyl group (hexadodecyl, C_16_) are conjugated to HA (HAC_16_), which are then able to self-assemble [34]. HAC_16_ micelles have been studied for ocular diseases, because they can interact with mucin resulting in an optimal mucoadhesion. Several lipophilic ophthalmic drugs, specifically corticosteroids, such as triamcinolone acetonide, triamcinolone, and dexamethasone, are loaded into the HAC_16_ micelles. Zhang et al. studied the increasing hydrophobicity of redox-sensitive PMs composed of HA and deoxycholic acid (DOCA) conjugated via disulfide bonds (HA-ss-DOCA) [9] (Figure 4A). Additionally, the targeted intracellular drug delivery of PTX loaded into HA-ss-DOCA with high entrapment efficiency (93.2%) and drug-loading capacities (34.1%) was studied. At high GSH levels, HA-ss-DOCA was reduced and disassembled to release drugs.

Polymer-drug conjugates (PDCs), which are composed of hydrophobic drugs conjugated with polymers, have diverse advantages, such as high drug loading and lower pre-drug leakage [35]. Additionally, PDCs are amphiphilic and can self-assemble into micelles. PTX was conjugated to HA in several ways, such as through disulfide bonds [36], dendrimers [37], and nanocomplexes (Figure 4) [38,39]. The Huo group presented HA-PTX conjugated micelles with redox-cleavable disulfide bond (HA-ss-PTX) [36]. The drug loading was over 40%, which is greater than other loading systems, and the redox sensitive micelles showed burst drug release leading to enhanced cytotoxicity and higher apoptosis. Additionally, Haag et al. disclosed endosomal pH-responsive prodrug PTX-conjugated HA dendritic oligoglycerol (HA-dOG-PTX) micelles [37]. With multivalent drug conjugation of dendrimer, it gives high drug-loading resulting in excellent cytotoxicity to human breast cancer xenografts. PTX is conjugated to HA through acetal groups, which are cleavable in endo/lysosomal pH and quickly released from the dendrimer. Thus, by adding hydrophobic small molecules, such as long alkyl chains and drugs, easy conjugation, amphiphilic properties, and high drug loading efficiency can be achieved for enhanced targeted anticancer therapy.

### 2.3. HA Coating Micelle

HA is a polyanionic biomacromolecule, which forms electrostatic interactions with positive nanocarriers. Coating HA on the surface of positive carriers provides active targeting to CD44 receptors and protects the carriers from drug leakage [39]. The Ryu group reported amphiphilic HA micelles with small molecules, such as mitochondria targeting NIR cyanine and peptide-drug conjugates, for selective cancer therapy [40,41]. They synthesized mitochondria targeting water-soluble indocyanine dye (IR-Pyr) with a positively charged pyridinium moiety and formed HA micelles (HA-IR-Pyr) via electrostatic interactions. The indocyanine dyes are sensitizers of photodynamic therapy (PDT); thus, the HA-IR-Pyr micelles provide greater anticancer therapy by targeting the mitochondria of cancer cells with PDT (Figure 5). HA-IR-Pyr showed good tumor accumulation with the SCC-7 tumor model compared to other PDT dyes indicating that HA is important to targeting tumor. Positively charged tripeptides (KCK-CPT) conjugated with CPT form nanotube structures in aqueous solution. The nanostructures are transformed to micelles (HA-KCK-CPT) via electrostatic interaction. HA-KCK-CPT provided higher selectivity to cancer cells (SCC-7 cells) compared to KCK-CPT. Liu et al. showed that GSH-responsive prodrug micelles consisted of amphiphilic ferrocenium-tetradecyl (Fe-C_14_), HA, and Dox [42]. In the GSH-rich tumor micro-environment, HA-Fe-C_14_/Dox micelles were reduced from hydrophilic ferrocene to hydrophobic ferrocene, resulting in dissociation of micelles and release of Dox. In vivo experiments showed that HA-Fe-C_14_/Dox has excellent inhibition of tumor growth. Wu et al. generated a double layered HA-coated PEI (polyethyleneimine)-*g*-stearic acid (PgS) that encapsulated a chiral drug, (−)-gossypol, for cancer therapy [43]. Hydrophobic stearic acid was grafted to PEI to make amphiphilic branched polymer (PgS). PgS can self-assemble into micelles in water and encapsulate (−)-gossypol ((−)-G) via electrostatic interaction, i.e., negatively charged HA covered positive charged PgS micelles. Double layered (−)-G-PgSHAs enhanced the stability of the chiral drug and the antitumor therapeutic effect in vitro and in vivo. The Luo group developed HA-coated PMs composed of poly(caprolactone)-*b*-poly(*N*,*N*-diethylaminomethyl methacrylate)-r-poly (2-(methacryloyloxy)ethyl phosphorylcholine (PCL-PDEAMPC) micelles, which is an amphiphilic co-polymer [44]. After HA coating, the micelle size slightly increased from 72.6 nm to 76.3 nm. Dox was loaded into the PCL-PDEAMPC micelle. Micelles themselves are not toxic to cells; however, Dox-loaded PCL-PDEAMPC-HA micelles are very toxic to 4T1 cells after a 48 h incubation. Therefore, HA-coated micelles provide several advantages, such as the safe delivery of hydrophobic drugs in carriers, superior cellular uptake, and antitumor efficacy. Characteristics of HA micelle nanomedicines are summarized in Table 1, including size and molecular length of HA, etc.

### 2.4. Conclusion of HA Micelles

Micelles are a class of nanocarrier widely used drug delivery systems or cancer therapy. Their nanometer-sized characteristics can provide effective anticancer therapy through the EPR effect. There are several advantages in using HA micelles for cancer therapy. Facile conjugation of HA with hydrophobic drugs or molecules can give amphiphilicity which can assemble into micelles by encapsulating drugs via chemical conjugation, physical absorption and electrostatic interaction. Additionally, negatively charged HA easily can cover the positively charged micelles resulting in enhanced selectivity via avoiding noncontrollable cellular uptake to normal cells and improved antitumor efficacy via releasing the drugs after specific stimuli, such as redox reactions and an acidic pH. Thus, HA micelles can become important drug delivery vehicles for targeted cancer cells. 

## 3. HA Nanogels

Nanogels are 3-dimensional hydrogels formed by physical or chemical crosslinking of polymers. Nanogels have a high affinity to water molecules due to their hydrophilic moieties. Despite that, nanogels are not dissolved in water but swell, which is a unique property of the nanogel caused by the crosslinking within polymers. Thus, nanogels have both characteristics of hydrogels and nanoparticles [45]. In addition, nanogels are attractive nanoplatforms because of many advantages, such as good biocompatibility, high colloidal stability, high drug loading capacity from large interior space, and tunable inner network. For these reasons, many types of crosslinking strategies using thiol groups [46], tetrabutylammonium (TBA) ions [47], cholanic acid [7], or graphenes have been widely studied to form HA-based nanogels (Figure 6) [48].

### 3.1. Methacrylate-Modified HA Nanogels

Methacrylate groups are frequently used to form HA-based nanogels with hydrophobic molecules. Zhong et al. reported an HA nanogel for targeted protein delivery (Figure 7). The group prepared the HA nanogels by inverse nanoprecipitation of methacrylate- and tetrazole-conjugated HA and the subsequent photo-crosslinking under UV light. Two intracellular apoptotic protein drugs, cytochrome c (CC) and granzyme B (GrB), were loaded into the nanogels. HA nanogels encapsulated the proteins safely due to photo-crosslinking and released nearly 99% of CC under high concentrations of GSH, which can degrade disulfide linkers in nanogels. They observed strong fluorescence of tetrazole in the cytoplasm in confocal laser scanning microscopy (CLSM) images, indicating high cellular internalization of HA nanogels into MCF-7 cells, which have an abundance of HA-receptors on their membrane. 

Inhibition of tumor growth using HA nanogels was also confirmed by an in vivo antitumor experiment [49]. In other studies by Zhong et al. the introduction of the GE11 peptide (YHWYGYTPQNVI) on the HA backbone allowed the nanogel to target epidermal growth factor receptor (EGFR); thus, the nanogel system was able to dual-target EGFR and CD44 over-expressed cells [50].

Jiang et al. reported enzyme-sensitive HA nanogels to target cancer. Methacrylate modified on HA binds with di(ethylene glycol) diacrylate (DEGDA), which act as a crosslinker between polymers, forming nanogels. Dox was encapsulated within the nanogel via diffusion into networks. The group determined that HA nanogels could preferentially internalize CD44-overexpressed 2D and 3D multicellular spheroids (MCs). The biodegradable ester linkage in the nanogels is cleaved by lipase and HAdase; thus, Dox molecules encapsulated in the nanogel can be released in the presence of enzyme. As Dox-loaded nanogels were treated with multiple enzymes (lipase and HAdase) and exposed to low pH condition, 75% of the encapsulated Dox was released within 120 h, indicating the degradation of the nanogels. Additionally, HA nanogels showed high penetration and accumulation at CD44 positive cell lines comparable to cell lines with fewer CD44 receptors [51]. Jiang et al. also used HA nanogels to selectively treat CD168 overexpressed cancer cells. Methacrylate-modified HA formed nanogels with the addition of a disulfide containing crosslinker; as a result, the nanogels targeted the CD168-expressed cell line and released Dox in the presence of GSH in cancer cells [52]. Tang et al. encapsulated the Dox in the methacrylate-crosslinked HA nanogels for targeted drug delivery. HA was functionalized with methacrylate group and further crosslinked using a di-methacrylate containing linker, forming nanogels. Through CD44-mediated endocytosis, loaded Dox was selectively delivered into the cancer cell [53].

### 3.2. Cholesterol-HA Nanogels

Cholesterol is a molecule which enables the construction of the cellular membrane. Modifications with cholesterol on HA not only endows hydrophobicity to HA, forming well-crosslinked nanogels, but also enhances the permeability of nanogels into the cell [54].

Vinogradov et al. demonstrated that HA-based nanogels conjugated with drugs could deliver cytotoxic nucleosides selectively to CD44-positive drug-resistant cancer cells (Figure 8). Cholesterol- modified HA (CHA) has been conjugated with etoposide (ETO), salinomycin (SAL), and curcumin (CUR) (CHA-ETO, CHA-SAL, and CHA-CUR, respectively). Each drug-conjugated CHA easily formed a 30 nm-sized nanogel after ultrasonication. Due to their hydrophobic moieties, cholesterol makes nanogels more compact. To improve the binding efficacy of HA to the CD44 receptor, HA with molecular weight of 60 kDa was selected so that nanogels with smaller size and high loading amount could be synthesized. While the UV absorbance of free CUR molecules decreased, CHA-CUR nanogels showed no significant decrease in absorbance within 24 h, indicating that the drug stability is greatly enhanced after loading in nanogels. In addition, CHA-drug nanogels had high cytotoxicity and cellular uptake in well-known drug-resistant human breast carcinoma cells, MDA-MB-231/F, and in pancreatic adenocarcinoma cells, MIA PaCa-2, compared with free drug molecules and nanogels without cholesterol [55]. Vinogradov et al. also reported that cholesteryl-HA can deliver CUR with high cellular permeability and accumulation. These nanogels efficiently killed cancer cells by inducing apoptosis and simultaneously suppressing the immune response mediators, such as nuclear factor-kappa B (NF-ĸB), tumor neurosis factor (TNF)-α, and chitosan oligosaccharides (COS)-2 [56]. Akiyoshi et al. designed cholesterol-containing HA nanogels for therapeutic delivery of the recombinant human growth hormone (rhGH) protein. In their nanogel system, size or density was controlled by adjusting the amount of salt, and rhGH was released in sustained manner [57].

### 3.3. Acetylated HA Nanogels

The hydrophobic acetyl group is generally used to form HA-based nanogels. Na et al. developed a low molecular weight acetylated HA for efficient drug delivery to cancer cells (Figure 9). As a strategy to impart hydrophobicity to HA to form nanogels, acetylation was performed on the HA backbone first, and Dox-loaded nanogels were prepared through a dialysis method. To investigate the effect of hydrophobicity on nanogels, different numbers of acetyl groups were modified on the HA (0.8, 2.1, and 2.6). Hydrophobic interactions, not only within the acetyl groups on the HA, but also between acetyl group and Dox molecules, reduced the swelling of the nanogels; this resulted in a nanogel size of less than 300 nm and enhanced loading efficiency (>90%) with higher degree of acetylation. Additionally, the amount of drug released in 1 day was 38%, 42%, and 67% for modifications to 0.8, 2.1, and 2.6 acetyl groups, respectively. This suggests that the drugs in the nanogels were stable. One of the merits of HA-based nanogels is biocompatibility. In a cell viability test, the survival rate of cells treated with nanogels only was above 90%; however, Dox-loaded nanogels showed significant cytotoxicity against HeLa cells which was comparable to free Dox molecules [58].

Na et al. reported an acetylated HA conjugated with a photosensitizer for PDT (Figure 10). HA was functionalized with acetic anhydride to form nanogels with pheophorbide (Pba), a highly singlet oxygen generating photosensitizer. Pba-quenching hydrogels can be degraded by enzymes in the endosome, and UV light-activated Pbas form singlet oxygen molecules, leading to intracellular dysfunction. The self-quenching effect due to the acetyl group on HA confirmed that the fluorescence intensity of pyrene decreased dramatically as the concentration of Ac-HA-Pba increased. Comparing the fluorescence of Ac-HA-Pba in DMSO and PBS, PBS prevented the interactions between the nanogels and the Pbas, leading to the self-quenching effect on Pbas. Ac-HA-Pba also showed high cellular uptake into HeLa cells, which is a CD44-overexpressed cell line, in CLSM images. Using fluorescent probes (CalceinAM and EthD-1), visible amounts of dead cells were observed by phototoxicity-induced UV irradiation [59]. In additional studies from Na et al., an acetylated HA-based nanogel was used as a drug carrier for PDT. Both triggers—light and acidic pH—induced the apoptosis of cancer cells [60]. We summarized characteristics of HA nanogel nanomedicines in Table 2, including size and molecular length of HA.

### 3.4. Conclusion of HA Nanogels

Nanogels are hydrogels physically or chemically crosslinked by polymers. While nanogels are water-friendly due to their hydrophilic moieties, crosslinked polymers make nanogels not fully dissolve in water. Nanogels can be constructed with various types of materials. Among them, hyaluronic acid is one of materials that can easily form nanogel simply by conjugating with hydrophobic group, such as methacrylate group, cholesterol, or acetyl group, et cetera. Further crosslinking within the materials is a facile method to build final nanogels. Hyaluronic acid is known for its high biocompatibility, so when it becomes nanogel, high drug-loading nanocarriers even with colloidal stability can be applicable to safe drug delivery system.

## 4. HA Inorganic Nanomedicines

Inorganic nanomedicines are attracting much attention due to the unique physicochemical properties of inorganic moieties. These properties include inertness, stability, and ease of functionalization [61,62,63]. HA modifications to inorganic nanoparticles not only increases the cancer targeting ability of inorganic nanomedicine [64], but also increases the theranostic effect, such as controlling drug release [65,66], colloidal stability [67], and increasing biocompatibility [68,69], of the nanomedicine. Inorganic particles that are primarily used in nanomedicine, including silica nanoparticles (SiNPs), gold nanoparticle (AuNP), and metal organic frameworks (MOF), have been actively studied in recent years.

### 4.1. SiNPs

SiNPs have excellent biocompatibility, systemic stability, tunable particle size, and controllable surface chemistry [70,71]. Mesoporous silica nanoparticles (MSNs), which are used for the delivery of biomolecules and cancer drugs to cancer cells, have advantages as drug carriers because of their high surface area and tunable pore size [64]. Hou et al. showed that 5-fluorouracil-loaded HA conjugated SiNPs (5-FU/HSNP) increased the therapeutic effect of colon cancer in vivo [69]. In order to modify HA on SiNPs, the surface functional group of the nanoparticle was converted from hydroxy (OH) to amine (NH_2_) by aminosilane (3-aminopropyltriethoxysilane (APTES)) (Figure 11). After modification, HA was conjugated on the surface of SiNPs. The amine functional group of the SiNPs was reacted with the carboxylic group of HA by an EDC reaction. 5-FU/HSNP prevented the rapid clearance of 5-FU, and thus had a better colon cancer therapeutic effect by inducing the cellular uptake of SiNPs.

Ding et al. used a different method for HA modification to the MSNs (Figure 12A) [72]. This method utilized electrostatic interactions between the negatively charged HA and positively charged MSNs which was modified by APTES to endow the positively charge. HA improved cancer cell targeting as well as increased cellular uptake, biocompatibility, and colloidal stability (Figure 12B). HA also triggered the controlled drug release. In a study by Qu et al., HA prevented unexpected drug release by blocking the silica pore [65]. The blocked pore was opened by HAdase, which is overexpressed in cancer cells. Thus, HA mediated the controlled drug release at cancer cells (Figure 12C). Siling et al. inserted the disulfide bond between the HA and silica conjugation for the dual stimuli responsive system [73,74]. 

The theranostic application of HA-modified SiNPs or MSNs are determined by the loaded drug in the nanoparticles. Cancer drugs, such as Dox [64,65], CPT [75], and curcumin, are loaded for chemotherapy [71], and photosensitizer, such as porphyritic derivates, are loaded for PDT to nanoparticles [76].

Monduzzi et al. prepared nanoparticles modified with different lengths of HA and compared their hydrodynamic size, zeta potential, and cellular uptake ability after 24 h. Lengths of each HA were 8–15 kDa (HA_S_), 30–50 kDa (HA_M_), and 90–130 kDa (HA_L_) and all of HA were modified on the surface of amine-coated SiNP via EDC coupling. Nanoparticle with longer chain of HA showed decrease in zeta potential. However, HA_L_-modified SiNP was the smallest in size as 168 ± 11 nm and size of HA_S_- and HA_M_-SiNP were 192 ± 15 nm, and 197 ± 8 nm each. This increasing tendency in size is because HA_L_ could interact with nanoparticle with multi-point attachment. In cell incubation experiment in HeLa cells, high cellular uptake was observed for HA_S_ modified SiNP [77]. Characteristics of HA silica nanomedicines are also summarized in Table 3.

### 4.2. AuNPs

AuNPs (gold nanoparticles, gold nanorods, gold star, and gold nanocages) are used in biomedical applications as biocompatible carriers for drug delivery [78], as contrast agents for diagnostic imaging [79], and as PTT agents [80]. AuNPs have many useful properties, such as biocompatibility, simple synthesis, easy surface modification, versatile conjugation with biomolecules, and tunable optical properties, for use in cancer theranostics [78]. AuNPs with strong surface plasmon resonance (SPR) absorption have great advantages in photothermal chemotherapy applications [81]. Their optical properties and light absorption range are controlled by the size and shape of the AuNPs [82,83]. The AuNPs (gold nanoparticles, gold nanorods, and gold stars) are nonporous nanoparticles, but gold nanocages are not. Therefore, drug modifications on the surface of the nanoparticle is a key element in the field of drug delivery [84]. HA is one of the useful backbones that allows modification of drugs and ligands to AuNP surface [85]. The carboxylic acid group of HA was functionalized to the sulfide or catechol group through HA modification for immobilization to the AuNP surface; that is, the functionalized HA forms an Au-sulfur bond or Au-catechol bond with AuNPs (Figure 11). 

Park et al. presented HHAuNP, which consists of HA, HiLytefluor^Tm^647 (Hilyte-647), and AuNPs for cancer diagnosis [79]. End thiol-functionalized HA labeled with Hilyte-647 fluorescence dye (HH) was modified to AuNPs by an Au-sulfide bond. This HH ligand lost the fluorescence due to the nanoparticle surface energy transfer (NSET) effect between the Hilyte-647 and AuNPs. When the modified HA was cleaved by reactive oxygen or HAdase, Hilyte-647 was released from AuNPs and fluorescence was recovered (Figure 13B).

Yao et al. and Xu et al. showed the dual targeting AuNPs, which have extra cancer targeting ligand (RGD, folate, anti-HER2 antibody) to enhance the cellular uptake of AuNPs [81,86,87]. Yao et al. showed the GNR-HA-^ALA/Cy7.5^-HER2, which is an HA, 5-aminolevulinic acid (ALA), Cy7.5, and anti-HER2 antibody modified to the gold nanorods (GNRs) [86]. Photosensitizers, ALA and fluorescent dye Cy7.5-NHS ester, were conjugated onto the GNR-HA via hydrazine and amide linkages, respectively. GNR-HA-^ALA/Cy7.5^-HER2 could significantly enhance the cellular uptake and facilitate the imaging (Cy7.5), PDT (ALA), and PTT (GNRs) (Figure 13A).

HA-AuNP are used in chemotherapy, PTT, PDT, and diagnostics [79]. For chemotherapy or PDT, drugs, such as Dox [80] and curcumin [85], and photosensitizers, such as porphyrin, are modified [88]. The stimuli-responsive release of the drugs facilitates specific cancer targeting and diagnosis. GNR is mainly used for PTT, because the GNR absorbs NIR region light that has high penetration ability [81,85,86,87]. The HA-GNR system enables PTT by generating heat without further modifications and facilitating controlled drug release (Figure 13C) [89,90,91]. Characteristics of HA gold nanomedicines are summarized in Table 4, including size and molecular length of HA.

### 4.3. MOF Nanoparticle

MOFs, composed of metal ions and organic linkers, have been studied for a variety of applications. Recently, these materials have been scaled down to nanometer sizes. Nanoscale MOFs (NMOFs) have been widely investigated for potential biomedical applications, such as drug delivery, imaging, PDT, and PTT, because of their high surface area, tunable shapes, and pore size properties, and controllable surface functionalities [92]. MOF can load a large amount of drug compared with other porous particles because of the high surface area and pore [93]. The coordination bonding of carboxylic acid with metal are used for HA conjugation on the MOF surface (Figure 11).

Zhao et al. showed that HA-coated Dox was loaded into zeolitic imidazolate framework-8 (Dox@ZIF-HA) for chemotherapy and MRI imaging [94]. ZIF-8, which is used as a nanocarrier, is easily degraded in tumor pH conditions, which promotes drug release [95,96,97]. To prepare the Dox@ZIF-HA, Dox@ZIF-8 was coated with polydopamine, successively chelated with Fe^3+^, and conjugated with HA by a coordination bond. Furthermore, the prepared Dox@ZIF-HA is a good contrast agent for magnetic resonance (MR) imaging because of the chelation of Fe^3+^ by PDA. Dox@ZIF-HA provides targeted drug delivery, stimuli responsive drug release, and cancer imaging.

HA-MOF nanomedicine enables chemotherapy and PTT by loading various drugs, such as Dox, indocyanine green, and cytarabine. HA-ZIF-8 is easily degraded in tumor pH conditions, which promotes the release of the drug [94,97]. Mil-100 (Fe), which has good T2-weighted MRI properties, is a drug carrier for cancer imaging (Figure 14) [98]. Porphyritic MOF enables PDT without additional compound loading, because the organic linker constituting MOF structure is used as a photosensitizer [99]. MOF has unique characteristics for each type and can be used for various cancer treatment applications to induce the synergistic effect with HA. Characteristics of HA MOF nanomedicines are summarized in Table 5, including size and molecular length of HA.

### 4.4. Conclusion of HA Inorganic Nanomedicines

HA is modified on various nanoparticles with many different methods. Silica nanoparticles are modified with HA by forming amide bonds between the amine groups on the silica surface and the carboxylic acid groups of HA or by electrostatic interaction. HA modification of gold nanoparticles is performed with post-functionalization of sulfide or catechol groups on HA. In the case of MOF nanoparticles, the metal ion in their structure coordinates with HA. Modification with HA endows nanoparticles with cancer targeting ability, enhances the colloidal stability and biocompatibility, and even functions as a gatekeeper to regulate the drug release. Introduction of HA onto inorganic nanoparticles gives improved cancer therapeutic effects.

## 5. Conclusions

In summary, HA micelles, nanogels, and inorganic nanomedicines can effectively transfer drugs to cancer cells through passive EPR and active targeting (receptor mediated internalization). The carboxylic acid groups of HA enable various modifications by simple reactions. HA expands the utilization of nanomedicines by increasing the biocompatibility, blood circulation time, cellular uptake, and colloidal stability of inorganic nanoparticles. In particular, various drugs can be loaded into the HA nanomedicine, facilitating chemotherapy, PTT, PDT, MRI, and fluorescence imaging. 

Summing up, we have reviewed the recent progress in the field of stimuli-responsive HA nanomedicines for cancer therapy and highlighted representative examples regarding the stimuli-responsive temporal and spatial drug delivery and release. The potential and progress of stimuli-responsive HA nanomedicines have been clearly shown in previous studies, yet, a correlation between HA and protein corona formation is still not established clearly. HA prevented the formation of protein corona on nanoparticles in one study [100], while it had no significant effect on inhibition of protein corona formation in another study [101]. Further studies on HA’s effect on protein corona formation need to be established. In the near future, we believe that the stimuli-responsive HA nanomedicines will become an important component of clinical cancer therapy.

## Figures and Tables

**Figure 1 pharmaceutics-11-00301-f001:**
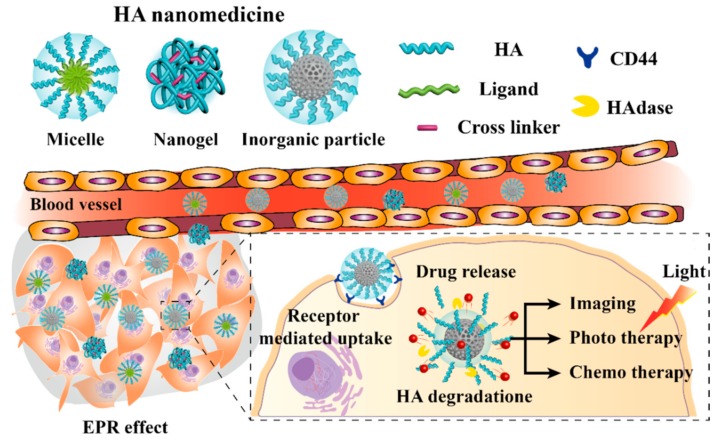
Schematic representation of the advantages and applications of hyaluronic acid (HA). EPR: Enhanced Permeability and Retention; CD44: Cluster Determinant 44.

**Figure 2 pharmaceutics-11-00301-f002:**
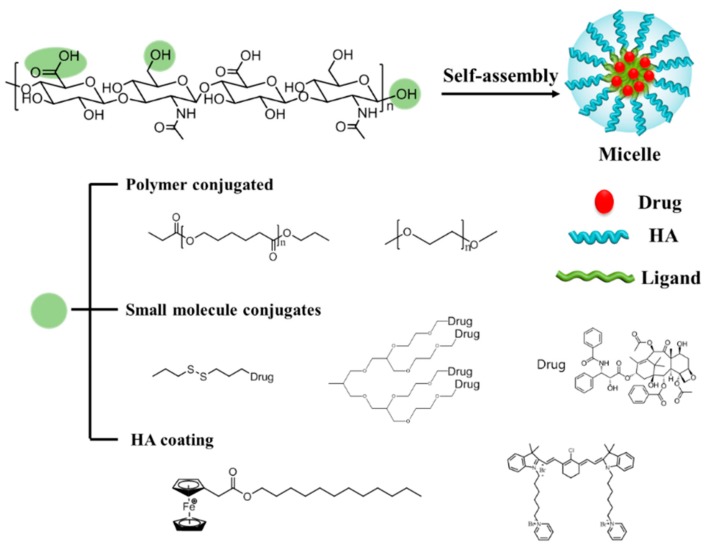
Schematic representation of hyaluronic acid (HA) polymeric micelles. Three possible sites for modification are shown as green circles. In aqueous solution, amphiphilic HA-ligands can self-assemble into micelles by encapsulating drugs or conjugating with drugs. Polymers can be grafted to the HA backbone. Block-co-polymers and small molecules, such as drugs and dendrimers, can be conjugated with HA. Positively charged molecules or polymers can form micelles with HA via electrostatic interactions.

**Figure 3 pharmaceutics-11-00301-f003:**
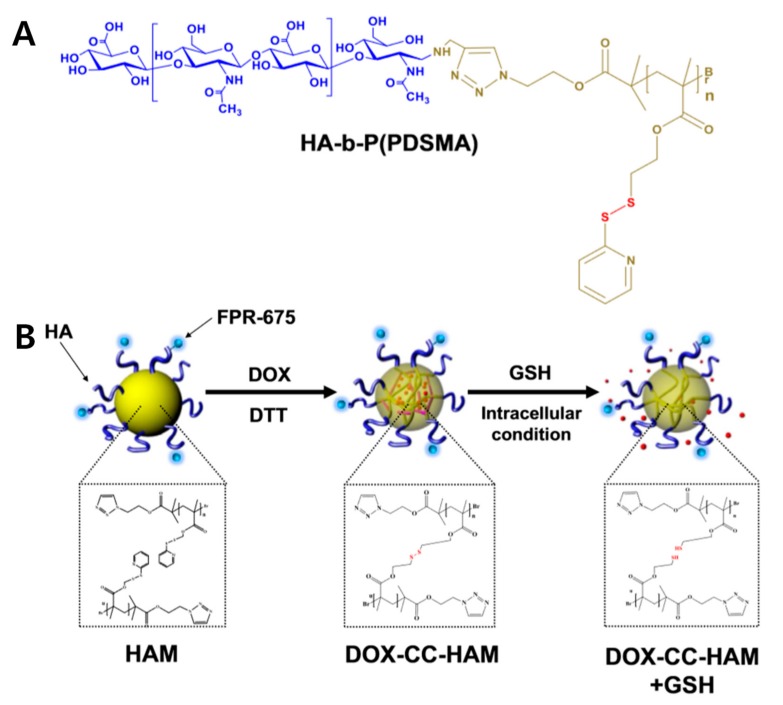
Redox-responsive hydrophobic polymer grafted hyaluronic acid (HA) polymeric micelles. (**A**) Chemical structure of HA-poly(pyridyl disulfide methacrylate (HA-*b*-P(PDSMA)), and (**B**) schematic illustration of crosslinked HA polymeric micelles with doxorubicin (Dox). GSH: Glutathione; CC-HAM; core-crosslinked polymeric micelles based on HA. Reproduced from [27] with permission. Copyright 2015 Elsevier Ltd.

**Figure 4 pharmaceutics-11-00301-f004:**
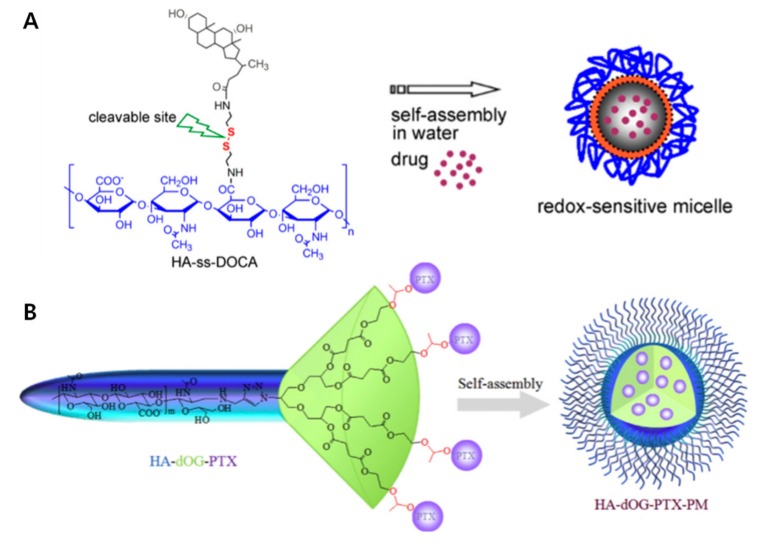
Small molecule conjugated hyaluronic acid (HA) micelles (**A**) Illustration and chemical structure of bio-reducible HA polymer and deoxycholic acid (DOCA) conjugated via disulfide bonds (HA-ss-DOCA) encapsulating drugs. Reproduced with permission [9] Copyright 2011 Elsevier Ltd. (**B**) PTX-conjugated HA dendritic oligoglycerol (HA-dOG-PTX) via an acetal group linker is responsive to acidic pH. Reproduced from [37] with permission. Copyright from 2016 Elsevier Ltd.

**Figure 5 pharmaceutics-11-00301-f005:**
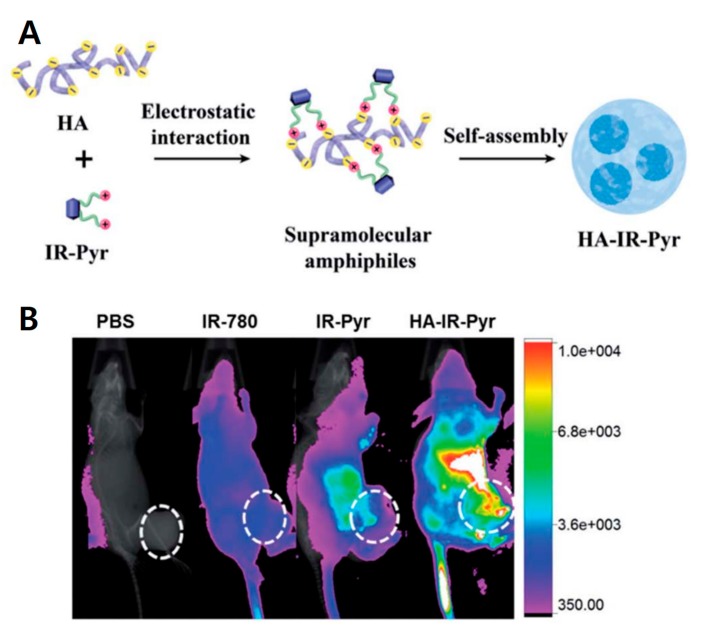
Schematic representation of hyaluronic acid (HA)-coated carriers for cancer therapy (**A**) HA-indocyanine dye (IR-Pyr) micelle formation scheme. (**B**) The importance of HA for HA-IR-Pyr tumor accumulation; HA-IR-Pyr has better localization compared to IR-Pyr and IR-780 dyes. Reproduced from [40] with permission. Copyright 2017 The Royal Society of Chemistry.

**Figure 6 pharmaceutics-11-00301-f006:**
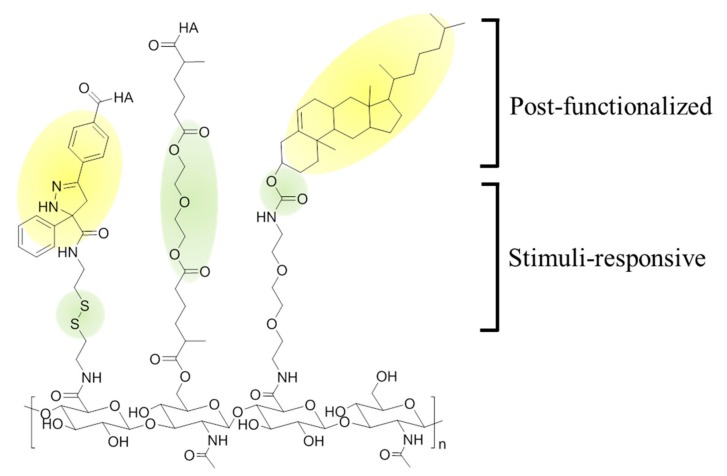
Chemical modification of hyaluronic acid (HA) with (1) functionality groups and (2) stimuli-responsive moieties. Photo-crosslinking between tetrazole and methacrylate generates a durable nanogel structure, while disulfide bonds endows degradability to nanogels under reductive condition (Left). A crosslinker functions as not only the precursor of the nanogel, but also enzyme-responsive moiety, such as ester group (Middle). To make the nanogels using hydrophilic HA, a hydrophobic moiety can be post-functionalized on the backbone. (i.e., cholesterol) (Right).

**Figure 7 pharmaceutics-11-00301-f007:**
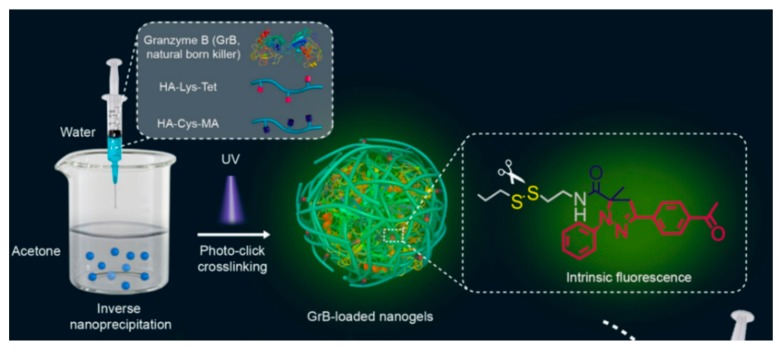
Schematic representation of a delivery platform with methacrylate- and tetrazole-modified hyaluronic acid (HA)-based nanogels for therapeutic proteins and granzyme B. Adapted from [49] with permission. Copyright (2016) American Chemical Society.

**Figure 8 pharmaceutics-11-00301-f008:**
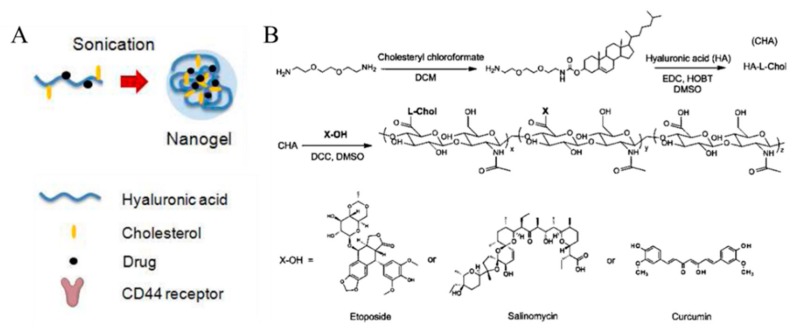
(**A**) Schematic representation of cholesterol-modified hyaluronic acid (HA) nanogel formation. (**B**) Scheme of modification of HA with cholesterol and drug molecules. Reproduced from [55] with permission. Copyright (2013) American Chemical Society. CHA: Cholesterol modified HA; DCM: Dichloromethane; EDC: 1-Ethyl-3-(3-dimethylaminopropyl)carbodiimide; HOBT: Hydroxy-benzotriazole; DMSO: Dimethylsulfoxide; DCC: *N*,*N*’-Dicyclohexylcarbodiimide.

**Figure 9 pharmaceutics-11-00301-f009:**
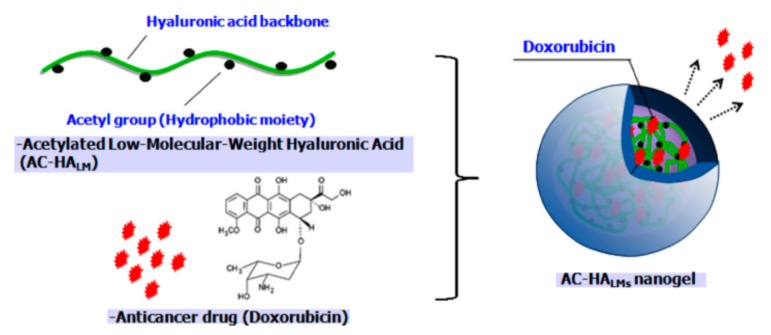
Schematic representation of anticancer drug, Dox, loaded into hyaluronic acid (HA) nanogels with conjugated acetyl groups. Reproduced from [58] with permission. Copyright 2010 from Elsevier Ltd.

**Figure 10 pharmaceutics-11-00301-f010:**
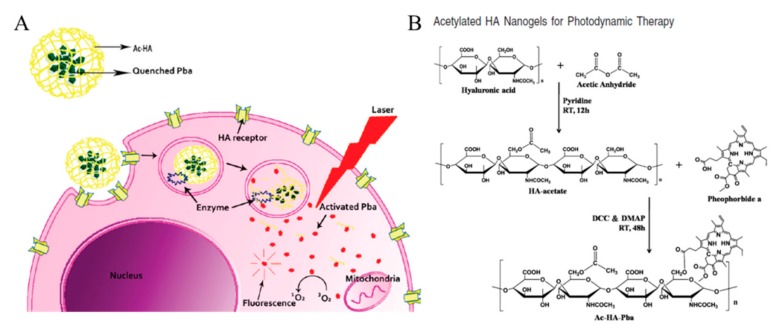
(**A**) Schematic representation of photosensitizer, pheophorbide (Pba), quenchable hyaluronic acid (HA)-based nanogels and single oxygen generation induced by Pba activation under UV irradiation. (**B**) Synthetic scheme of acetylated HA nanogels further conjugated with Pba. Reproduced from [59] with permission. Copyright (2010) American Chemical Society. DCC: *N*,*N*’-Dicyclohexylcarbodiimide; DMAP: 4-Dimethylaminopyridine.

**Figure 11 pharmaceutics-11-00301-f011:**
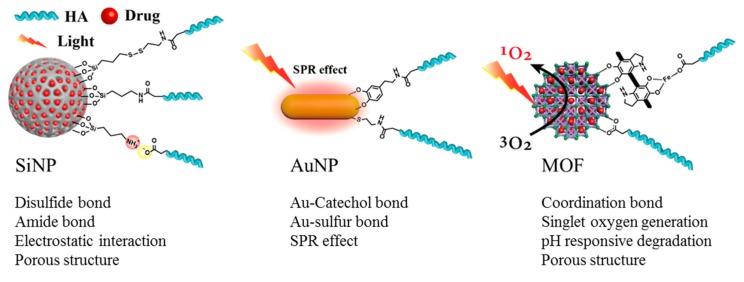
Schematic representation of inorganic nanoparticle hyaluronic acid (HA) modification methods and properties. SiNP: silica nanoparticle; AuNP: gold nanoparticle; MOF: metal organic frameworks; SPR: Surface plasmon resonance.

**Figure 12 pharmaceutics-11-00301-f012:**
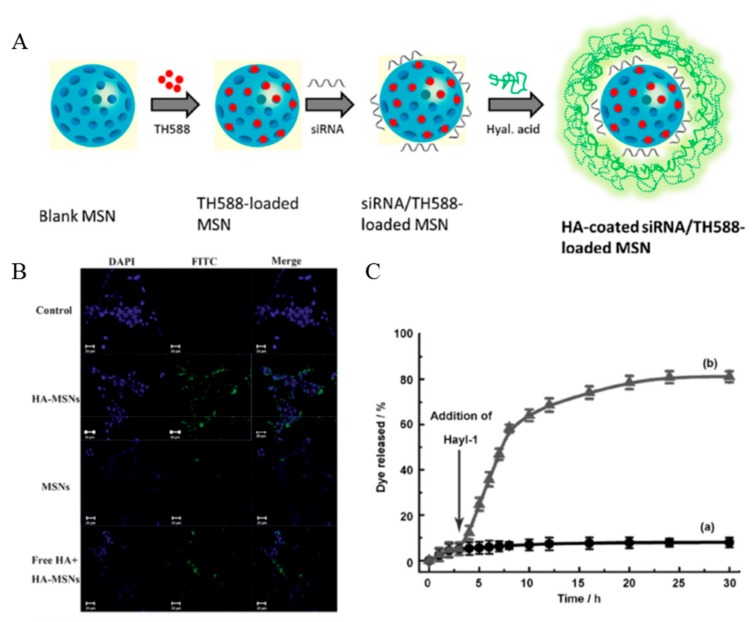
(**A**) Schematic representation of the preparation of TH287 and siRNA-loaded hyaluronic acid (HA) assembled mesoporous silica nanoparticles (MSNs). Reproduced with permission [72] Copyright 2018 Elsevier Ltd. (**B**) Confocal microscopy images of HCT-116 cells without any treatment as a control (first row), with the treatment fluorescein isothiocyanate (FITC) labelled HA-MSNs (second row), MSNs (third row), and free HA (10 mg mL^−1^) combined with FITC labelled HA-MSNs (last row). Reproduced with permission [64] Copyright 2013 The Royal Society of Chemistry. (**C**) Responsive release of rhodamine B from MSNs-HA by the addition of HAdase solution after incubation for 3 h. Reproduced from [65] with permission. Copyright 2013 John Wiley & Sons, Inc.

**Figure 13 pharmaceutics-11-00301-f013:**
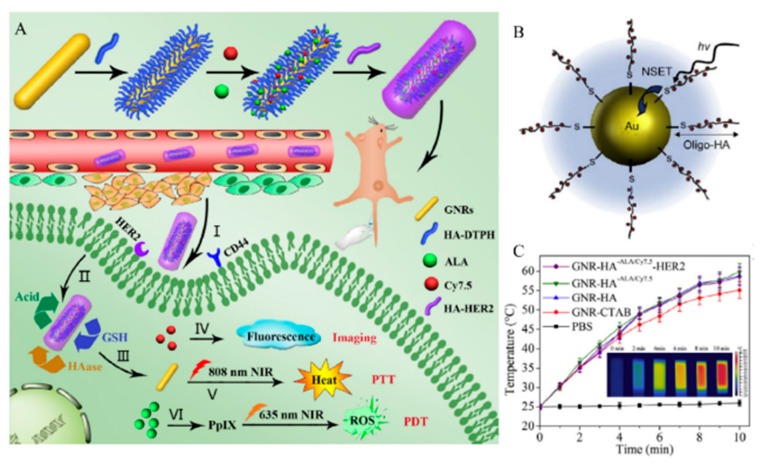
(**A**) Schematic representation of the processes for preparing hyaluronic acid (HA)-gold nanorods (GNR) with responsive drug release and its application. Reproduced with permission [87] Copyright 2018 Elsevier Ltd. (**B**) Schematic illustration of nanoparticle surface energy transfer (NSET) interactions between Hilyte-647 dye labeled oligo-HA and gold nanoparticles (AuNPs). Reproduced with permission [79] Copyright 2008 Elsevier Ltd. (**C**) Corresponding infrared thermal images of GNRs-HA-Folic acid (FA)-doxorubicin (DOX) suspension at different time intervals. Reproduced from [87] with permission. Copyright 2018 Elsevier Ltd.

**Figure 14 pharmaceutics-11-00301-f014:**
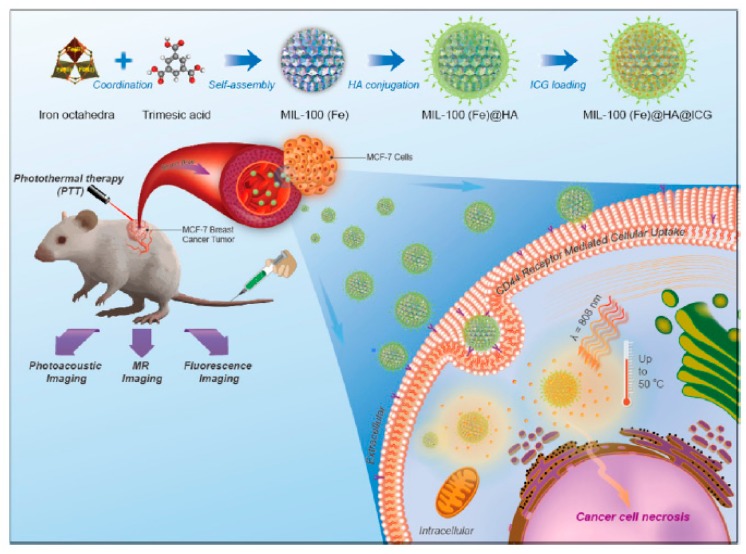
Schematic representation of the synthesis procedure, hyaluronic acid (HA) conjugation, Indocyanine green (ICG) loading, and multimodal imaging-guided photothermal therapy (PTT) of MIL-100(Fe) nanoparticles (NPs). Reproduced from [98] with permission. Copyright (2017) American Chemical Society.

**Table 1 pharmaceutics-11-00301-t001:** Characteristics of hyaluronic acid (HA) micelle nanomedicines.

Component	Cargo	Drug loading Content (%)	Responsive Stimuli	Therapy Method	Size (nm)	Surface Charge (mV)	HA MW (kDa)	Ref.
2-(Pyridine-2-yldisulfanyl)-ethyl methacrylate	Dox	8	GSH	Chemo	148–215		7.4	[27]
Poly(ethylene glycol)	CPTDox	32–34	HAdase	Chemo	320		234	[29]
β-Dendritic oligoglycerol	PTX	20.6	pH	Chemo	120	−21	9.5	[37]
Deoxycholic acid (DOCA)	PTX	34.1	GSH	Chemo	130–300	−31–−37	11	[9]
Hexadecylamine	Dexamethasone, triamcinolone, triamcinolone acetonide	2–6	HAdase	Chemo	190–−350	−15–−46	<10	[34]
Ferrocenium tetradecyl	Dox	5.97	GSH	Chemo	117	−25.7	100	[42]
Indocyanine dye	IR-Pyr		HAdase, ROS	PDT	30	−40		[40]
Peptide-drug	CPT		HAdase	Chemo	44	−9.4		[41]
Polyethylene-imine (PEI) -stearic acid	(−)-Gossypol		HAdase	Chemo	110.9	−29.6	>100	[43]

DOX: Doxorubicin; PTX: Paclitaxel; CPT: Camptothecin; IR-Pyr: Indocyanine dye; ROS: Reactive Oxygen Species; Chemo: Chemotherapy; GSH: Glutathione.

**Table 2 pharmaceutics-11-00301-t002:** Characteristics of hyaluronic acid (HA) nanogel nanomedicines.

Component	Cargo	Drug Loading Content (%)	Responsive Stimuli	Therapy Method	Size (nm)	Surface Charge (mV)	HA MW (kDa)	Ref.
HA-Cys-methacrylate, HA-Lys-tetrazole	Cytochrome C, granzyme B	89.2	GSH	Protein (Induce apoptosis)	150	−17.6	35	[49]
Methacrylated HA, DEGDA	Dox	16	Enzyme (lipase and HAdase)	Chemo	50	−45.0	7	[51]
Membrano-tropic cholesteryl-HA, cholesterol-modified HA	Etoposide, salinomycin, and curcumin	20	Hydrolysis (ester linkage)	Chemo	20–40	−31.6–−41.4	62	[55]
Methacrylated HA	Dox	24	GSH (Disulfide bond)	Chemo	79	−40	0.5	[58]
Cholesteryl-HA	Curcumin	20	pH, proteolytic enzyme	Chemo	29.2	−38.4	0.58	[59]
Acetic anhydride	Dox	93.1		Chemo	275	−31	7	[52]
Acetic anhydride	Pheophorbide	0.31 per 1 unit of HA	Enzyme	Photo-dynamic	125–150	−21–−34	62	[56]

GSH: Glutathione; Chemo: Chemotherapy; Dox: Doxorubicin.

**Table 3 pharmaceutics-11-00301-t003:** Characteristics of hyaluronic acid (HA) silica nanomedicines.

Bond	Pore	Cargo	Drug Loading Content (%)	Responsive Stimuli	Therapy Method	Size (nm)	Surface Charge (mV)	HA MW (kDa)	Ref.
Amide	O	Dox	1.2	X	Chemo	70–100	−18.8	200	[64]
Electrostatic interaction	O	TH287, MDR1 siRNA	8.91	X	Chemo	184	−18.4		[72]
Amide	O	6-Mercapto-purine		GSH	Chemo	80	−25.5–−27.9	10	[73]
Amide	O	CPT	0.96	X	Chemo	100	−14.9	18	[75]
Amide	O	Dox	0.01	HAdase	Chemo	232	−30	100	[65]
Amide	X	5-Fluoro-uracil	15	X	Chemo	138		35	[69]

GSH: Glutathione; Chemo: Chemotherapy; Dox: Doxorubicin; CPT: Camptothecin.

**Table 4 pharmaceutics-11-00301-t004:** Characteristics of hyaluronic acid (HA) gold nanomedicines.

Bond	Pore	Cargo	Drug Loading Content (%)	Responsive Stimuli	Therapy Method	Size (nm)	Surface Charge (mV)	HA MW (kDa)	Ref.
Au-sulfur.	X	Hilyte-647	X	HAdase, ROS	Diagnosis	16		1790	[79]
Au-catechol	O	Dox	X	HAdase, NIR	Chemo, PTT	50	−25.7	100	[80]
Au-catechol	X	Dox	X	pH, NIR	Chemo, PTT	71	−11.4	8	[81]
Au-sulfur	X	ALA, Cy7.5	X	HAdase, GSH, pH	PTT, PDT	72	−13.8	8	[87]

Dox: Doxorubicin, ALA: 5-aminolevulinic acid; NIR: Near infrared; PDT: Photodynamic Therapy; PTT: Photothermal Therapy.

**Table 5 pharmaceutics-11-00301-t005:** Characteristics of hyaluronic acid (HA) MOF nanomedicines

Bond	Pore	Cargo	Drug Loading Content (%)	Responsive Stimuli	Therapy Method	Size (nm)	Surface Charge (mV)	HA MW (kDa)	Ref.
Coordination	O	Dox		pH	Chemo, MRI	150	−30.2	X	[94]
	O	Indocyanine green	42	NIR	PTT, MRI, PA imaging	106	−25.4	X	[98]
	O	α-Cyano-4-hydroxy-cinnamate		HAdase, Light	PDT	152	−13.9	X	[99]
Coordination	O	Cytarabine, indocyanine green	39.8	pH	Chemo, PTT	135		10	[97]

Dox: Doxorubicin, MRI: magnetic resonance imaging; NIR: Near infrared; PDT: Photodynamic Therapy; PTT: Photothermal Therapy.

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
