# Peer review of "Hyaluronic Acid-Coated Nanomedicine for Targeted Cancer Therapy"

_pharmaceutics, 2019, doi:10.3390/pharmaceutics11070301_

Round 1
Reviewer 1 Report
This paper is an interesting overview of the use of hyaluronic acid in nanomedicine. The paper is well structured and easy to read.
I only have some of points that, in my opinion, deserve to be discussed/clarified in the text.
The authors have listed different HA-based nano-formulations to be used for cancer therapy. Once a nanoparticle is injected in the blood, protein are generally adsorbed on the external surface. This leads to the formation of a "protein corona" which can affect the final destiny of the nanomedicine. Please discuss this phenomenon for the different nanomedicines (micelles, nanogels, inorganic particles).
HA is a linear polymer which can be obtained with different range of molecular weight (chain length). What is MW range of HA in the reviewed papers?
Nairi et al. (Colloid and Surfaces B, Bionterfaces 2018, 168, 50) have found that HA chain length affects the CD-44 mediate internalization of MSN-HA particles in Hela cells. Please discuss.
About HA conjugated to mesoporous silica nanoparticles the authors called this nanomedicne: e.g. SiNP e MSN. Please use only one name for the same preparation.
at pag 13, lines 332-333 the authors say:"This method utilized electrostatic interactions between the negatively charged HA and positively charged SiNPs". In fact SiNPs are negatively charged, unless they have been functionalized with APTES (aminopropyltriethoxysilane). Please clarify this point.
Reviewer 2 Report
Dear author,
I have read the present review about Hyaluronic Acid-Coated Nanomedicine for Targeted Cancer Therapy by Kim et al with high interest. The article is interesting, descriptive and provide important information about using HA-coated nanomedicine for cancer therapy.
I recommend to publish the paper after minor revision.
1. The abstract is not convincing. I recommend to write few more lines to expand the abstract about the importance of HA towards the cancer tageting and therapy.
2. The author needs to add few lines about the advantages or disadvantages of each subsection of HA -coated nanomedicine. Each section needs some conclusion which is missing now.
3. Please write a short section about the challenges and future outlook of the HA coated nanomedicine towards cancer therapy.
Overall the article is well written. I recommend to accept the manuscript after these revisions.
Reviewer 3 Report
A nice review of the uses of hyaluronic acid in nanomedicine targeting cancer. Good coverage.
Round 2
Reviewer 1 Report
The authors have revised their manuscript in a satisfying manner. I suggest its acceptance.